# Crack-Based Sensor by Using the UV Curable Polyurethane-Acrylate Coated Film with V-Groove Arrays

**DOI:** 10.3390/mi14010062

**Published:** 2022-12-26

**Authors:** Jongsung Park, Dong-Su Kim, Youngsam Yoon, Arunkumar Shanmugasundaram, Dong-Weon Lee

**Affiliations:** 1Department of Precision Mechanical Engineering, Kyungpook National University, Sangju 37224, Republic of Korea; 2School of Mechanical Engineering, Chonnam National University, Gwangju 61186, Republic of Korea; 3Department of Electrical Engineering, Korea Military Academy, Seoul 01805, Republic of Korea; 4Advanced Medical Device Research Center for Cardiovascular Disease, Chonnam National University, Gwangju 61186, Republic of Korea

**Keywords:** polyurethane-acrylate (PUA), crack-based strain sensor, high sensitivity, excellent fidelity, human motion monitoring

## Abstract

Over the years, several bare metal and crack-based strain sensors have been proposed for various fields of science and technology. However, due to their low gauge factor, metal-based strain sensors have limited practical applications. The crack-based strain sensor, on the other hand, demonstrated excellent sensitivity and a high gauge factor. However, the crack-based strain sensor exhibited non-linear behavior at low strains, severely limiting its real-time applications. Generally, the crack-based strain sensors are fabricated by generating cracks by bending a polymer film on which a metal layer has been deposited with a constant curvature. However, the random formation of cracks produces nonlinear behavior in the crack sensors. To overcome the limitations of the current state of the art, we propose a V-groove-based metal strain sensor for human motion monitoring and Morse code generation. The V-groove crack-based strain sensor is fabricated on polyurethane acrylate (PUA) using the modified photolithography technique. During the procedure, a V-groove pattern formed on the surface of the sensor, and a uniform crack formed over the entire surface by concentrating stress along the groove. To improve the sensitivity and selectivity of the sensor, we generated the cracks in a controlled direction. The proposed strain sensor exhibited high sensitivity and excellent fidelity compared to the other reported metal strain sensors. The gauge factor of the proposed V-groove-induced crack sensor is 10-fold higher than the gauge factor of the reported metal strain sensors. In addition, the fabricated V-groove-based strain sensor exhibited rapid response and recovery times. The practical feasibility of the proposed V-groove-induced crack-based strain sensor is demonstrated through human motion monitoring and the generation of Morse code. The proposed V-groove crack sensor can detect multiple motions in a variety of human activities and is anticipated to be utilized in several applications due to its high durability and reproducibility.

## 1. Introduction

Wearable electronics is a rapidly developing field of study in human motion monitoring, artificial intelligence robots, and human-machine interaction applications [1,2,3]. Most of the wearable electronics have a metal-based strain sensor, especially for human motion monitoring [4,5,6], and a multi-function chemical sensor for electrolytic analysis of the chemical sensor. Thus far, there have been numerous wearable strain sensors reported for measuring the human signal, such as heart rate [7], blood pressure [8], and motion [9]. The existing wearable strain sensors have made effective use of highly elastic polymers to embed electrically active materials such as silver or gold nanowires (AgNW or AuNW) [10], carbon nanotubes (CNT) [11], graphene [12], graphite [13], and other materials, including inherently conducting polymers [14,15]. Wearable strain sensors, on the other hand, require high stretchability (>50%) and sensitivity (GF > 100) with a wide detection range. In addition, it has been proven difficult to achieve high sensitivity because large stretching requires structural integrity and significant changes in the electrically active area [16,17,18]. As a result, several studies in recent years have shown that piezoresistive structures can be used to create highly sensitive strain sensors [19,20]. 

The resistive strain sensors, formed by the combination of conductive metals and flexible substrates, have attracted extensive research interest for monitoring human motion [21,22] because of their easy fabrication and fast response mechanism. For human motion monitoring, the wearable strain sensors are designed based on two important parameters, namely sensitivity and sensing range [23,24]. In addition, monitoring both tiny changes (such as pulse and swallowing) and large strain changes (such as finger and joint movements) is also being considered for wearable strain sensors. However, the low gauge factor of metals provides less displacement and always has a trade-off between sensitivity and sensing range. However, there is always a trade-off between sensitivity and sensing range. Significant structural change with a small change in strain results in high sensitivity. In contrast, the wide sensing range requires the sensor to maintain a conductive path under large deformation.

Recently, a new biomimetics sensing mechanism with an effective design and high sensitivity for providing high sensing performances was developed. For example, the crack-based sensor possesses high sensitivity even for small displacement or voice vibration [25]. The bioinspired crack-based sensors apply a strain of about 2% to create micro- or nanoscale physical cracks in both electrical metals and flexible substrates [26,27]. For instance, Liu et al. proposed a new strategy to develop fiber shaped stretchable strain sensors with significantly improved sensitivity [28]. Chen et al. reported a highly sensitive and fully flexible vibration sensor with a channel-crack-designed suspended sensing membrane for high dynamic vibration and acceleration monitoring [29]. Jiang et al. proposed stretchable strain sensors based on auxetic mechanical metamaterials. The proposed crack sensor showed a 24-fold sensitivity improvement compared to conventional sensors [30]. A high-performance flexible strain sensor with bio-inspired crack arrays has been proposed for the detection of human motions and surface folding [27]. Li et al. proposed a bioinspired robot skin with mechanically gated electron channels for sliding tactile perception [31]. The crack sensor has a high gauge factor of between 11 and 2000 and can monitor even small vibrations or sound waves. Though the crack-based sensor has higher sensitivity compared to that of a piezoresistive sensor, it does not have a uniform primary resistance for each device as the number of cracks cannot be controlled. Furthermore, the majority of crack-based sensors operate within a 2% tension range, with the crack creating extending ranges on the sensor surface [32]. To overcome these drawbacks, researchers have developed advanced fabrication methods for the crack-based sensor and extended the application fields. Furthermore, special equipment was required to make the crack, and still, it did not have a uniform initial resistance and sensitivity.

In this study, we developed a V-groove-induced crack-based sensor with a simple fabrication technique for human motion monitoring and Morse code generation. Firstly, to control the initial resistance, we employed a V-groove arrayed shape by using a silicon wet etched mold that controls the number of cracks in the sensor. In this fabrication method, each sensor had the same number of cracks, and through them, the value of the resistance changed when the sensor deformed, and the initial resistance value of the sensor was able to be constantly adjusted. Imprinting a PDMS (polydimethylsiloxane) embossed pattern on a PET (polyethylene terephthalate) film coated with PUA (polyurethane-acrylate) could transfer the fabricated V-groove pattern. PUA is a UV-curable material, and it can be uniformly coated on a substrate using a process such as spin coating. These procedures can be carried out in conjunction with the existing MEMS processes, thereby providing the convenience of various processes. After the first procedure, which employs the V-groove arrayed shape, the metallization process was performed, fine cracks were formed using a tensile device, and the amount of change in the opening and closing of the crack was measured. We verified the possibility of monitoring human movement and various applications using the manufactured crack-based sensor, and we expect it to be used as a more stable wearable device.

## 2. Materials and Methods

### 2.1. Materials

A commercially available pure silicon wafer was purchased from 4science Co., Ltd., Seoul, Korea. The photoresists AZ-GXR 601-46cp AZ 300 MIF were purchased from K1-solution Co., Ltd., Gwangmyung, Gyeonggi, Korea. The tetramethylammonium hydroxide (TMAH) was purchased from 4science Co., Ltd., Seoul, Korea. The polydimethylsiloxane (PDMS) was purchased from Dowhitech Co., Ltd., Goyang, Gyeonggi, Korea. The polyethylene terephthalate (PET) and polyurethane acrylate (PUA) were purchased from i-one film Co., Ltd., Anyang, Gyeonggi, Korea and MCNET Co., Ltd., Gwangju, Gyeonggi, Korea, respectively.

### 2.2. Fabrication of the V-Groove-Induced Crack-Based Strain Sensor

The detailed fabrication process flow of the V-groove-induced crack-based strain sensor is schematically illustrated in Figure 1. The fabrication of the V-groove-induced crack-based strain sensor consists of four important steps: (i) fabrication of the V-groove master mold based on a Si wafer, (ii) first replica fabrication using PDMS, (iii) second replica fabrication using PUA, and (iv) depositing metal on the second replica and formation of cracks. The details of each fabrication step are described in detail in the following sections. Firstly, as shown in Figure 1a, a V-groove was formed on the Si wafer using bulk micromachining technology. In a typical fabrication process, a 500 nm thick silicon dioxide (SiO_2_) sacrificial layer was initially grown on a Si wafer using a thermal oxidation (wet oxidation) process. Then, AZ GXR-601 46-cp photoresist was exposed to ultraviolet (UV) light (9.6 mW cm^−2^) for 8 s through a mask aligner for 90 s. The AZ GXR-601 46cp was developed using AZ 300MIF developer for 1 min and rinsed with DI water for 1 min. Subsequently, the substrate was dried with nitrogen (N_2_) gas to define the line pattern. The upper SiO_2_ layer was selectively etched using a buffered oxide etchant (BOE), and the photoresist remaining on the surface was removed using acetone. Next, the Si wafer was placed in a TMAH bath, and the Si was selectively etched to form a V-groove pattern (Figure 1a). Subsequently, the remaining SiO_2_ on the surface of the Si wafer was removed using BOE. The PDMS mixture was prepared by mixing a base polymer and a curing agent in a 10:1 ratio. The mixture was kept in a vacuum desiccator for 10 min to remove air bubbles. The PDMS mixture was poured into a Si wafer master mold and cured for 2 h on a 100 °C hot plate. The V-groove-patterned PDMS film was peeled off the surface of the Si master mold after curing, as shown in Figure 1b,c. The PET film and PUA were used for the fabrication of the second replica. For ease of handling, the PET film was attached to a Si wafer coated with a thin layer of PDMS as a soft adhesive. Subsequently, 5 mL of PUA, a UV-curable resin, was placed on the substrate, as shown in Figure 1d. Then, the substrate was spin coated to form the PUA layer at 1000 rpm for 40 s. The first replica was placed on a PUA-coated PET substrate (125 μm thick PET), and the PUA solution was weakly cured by UV irradiation (12 mJ cm^−2^) (Figure 1e). After peeling off the PDMS 1st replica in the longitudinal direction of the V-groove, the PUA layer was exposed to UV light at an exposure energy of 20 mJ cm^−2^ for 10 h to complete the preparation for V-groove replication, as shown in Figure 1f. Finally, using an electron beam evaporator at a deposition rate of 1 nm s^−1^ and a chamber pressure of less than 5 10^−6^ torr, 30 nm of chromium (Cr) and 30 nm of gold (Au) layers were deposited on the V-groove pattern PUA substrate. The PUA film was stretched (2%) using a custom stretcher to realize the Cr/Au thin film-based crack sensor, as shown in Figure 1g.

### 2.3. V-Groove-Induced Crack Sensor Device Concept and Measurement Techniques

The proposed V-groove-based crack strain sensor is schematically illustrated in Figure 2. The V-groove-based crack strain sensor is fabricated using a modified photolithography technique. The V-groove-based crack strain sensor is composed of three layers, such as the bottom PET layer, the middle PUA layer with a V-groove pattern, and the top Cr/Au metal layers with micro- and nanogrooves (Figure 2a). The micro- and nanogrooves were formed on the top Cr/Au metal layers using a custom-designed tensile testing machine (Figure 2b). Tension and compression (0–2%) were applied to the Cr/Au metal layers deposited on the V-groove patterned PUA coated PET substrate to form micro- and nanocracks in the metal layers. The change in resistance of the V-groove based crack strain sensor was measured using a LabVIEW-based data acquisition system (PXI-4071, National Instruments Inc., Austin, TX, USA).

## 3. Results and Discussions

### Preliminary Characteristics of the V-Groove-Induced Crack-Based Strain Sensor

The morphology of the fabricated V-groove-induced crack-based strain sensor is investigated through field emission scanning electron microscopy (FESEM), as shown in Figure 3. The panoramic view confirms the formation of a V-groove pattern with uniform shape and size. The clearly defined, well-faceted, and sharp edges demonstrate the excellent fidelity of the V-groove-induced crack-based strain sensor (Figure 3a). The top view and cross-sectional view of the cracked V-groove-induced crack-based strain sensor under externally applied pressure are shown in Figure 3b,c. In addition, the higher magnification image demonstrates that the cracks are generated when the tensile stress is applied to the sensor substrate in the direction of the V-groove. 

The applied stress significantly influences the depth and width of the cracks in the V-groove-based strain sensor. As a result, we investigated the effect of bending stress on the resistance change of the V-groove-induced crack-based strain sensor at various applied displacements of 1, 2, and 3 mm, respectively. The change resistance of the sensor was measured at different bending angles such as 30°, 60°, and 90° as shown in Figure 4a–c. The initial resistance of the crack sensor increases with increasing bending angles. The base resistance of the crack sensor was found to be 50.4 ± 0.3, 65.3 ± 2.4, and 70.3 ± 7.2 Ω toward 30°, 60°, and 90°, respectively. The increase in base resistance of the crack sensor with increasing bending angle could be attributed to the formation of more depth and width of the cracks while applying the external bending stress. The formation of more cracks on the sensor decreased the contact point of the crack in the top metal layer, thus resulting in an increase in the initial resistance value. Although the base resistance of the crack sensor changed with respect to bending angles, the sensor greatly responded to the applied displacement, as shown in Figure 4a–c. The change in resistance of the crack sensor at 30°, 60°, and 90° to 3 mm of applied displacement was ~53.9 ± 1.1, 77.4 ± 2. 2, and 98.8 ± 2.4, respectively. The maximum change in resistance was observed at 90° due to the formation of more cracks in the V-groove-induced crack sensor. The repeatability of the change in resistance of the crack sensor was measured at a 90° bending angle toward 1, 2, and 3 mm, as shown in Figure 4d. The crack sensor showed an excellent linear relationship to the applied displacement, demonstrating the excellent reliability of the proposed V-groove-induced crack sensor. The gauge factor of the fabricated crack sensor was calculated from the slope of the linear fit, and it was found to be 131.15 (Figure 4d). The gauge factor of the proposed V-groove-induced crack sensor was 10-fold higher than the gauge factor of the metal strain sensor but 10-fold lower than the reported crack-based sensor [25]. However, the proposed V-groove-induced crack sensor exhibits a wide operating range (0–10%) and excellent reliability at the base resistance. The strain applied to the sensor was calculated using an equation reported in the previous manuscript [27] and described in the Appendix A.

The response and recovery time are critical factors in determining the feasibility of the proposed V-groove-induced crack sensor. The response and recovery time of the V-groove-induced crack sensor were measured by applying a displacement of 5 mm. The V-groove-induced crack sensor rapidly responded to the applied displacement. The applied displacement was maintained for 60 s and then withdrawn from the crack sensor, which measured the recovery time of the sensor. The recovery time of the sensor was found to be 150 s. However, the recovery time of the sensor was significantly decreased when the external force was withdrawn after a few seconds. These findings demonstrate that the sensor exhibits a slightly poor response when subjected to the prolonged application of a substantial external force. Therefore, in this study, we apply an external strain to the crack sensor and then remove it in a matter of seconds. We also assess the hysteresis of the proposed V-groove sensor under 1.5 mm of externally applied displacement loading and unloading, as shown in Figure 5b. The resistance of the crack sensor was gradually increased by applying the 1.5 mm displacement and recovered back to its original resistance once the external force was withdrawn from the crack sensor. The sensor showed 10% hysteresis while loading and unloading 1.5 mm of applied displacement. The hysteresis of the V-groove-induced crack sensor is attributable to the deformation of the polymeric substance (PUA) utilized as a substrate for the formation of the metal layer. The polymeric materials recover from deformation more slowly than other metal-based strain sensors. The repeatability and long-term durability of the V-groove-induced crack sensor were investigated at 3 Hz stimulation. The crack sensor’s resistance was measured for 500 s while being stimulated at 3 Hz. The change in resistance of the crack sensor when stimulated at 3 Hz was measured to be 56 Ω. Due to the polymeric material’s hysteresis and slow deformation recovery, the base resistance value increased slightly during continuous stimulation. Nonetheless, the sensor retains a consistent resistance change of 56 under continuous stimulation.

The practical applicability of the sensor for human motion monitoring was demonstrated by fixing the V-groove-induced sensor on the index finger, as shown in Figure 6a. The crack sensor mounted on the joint of the index finger caused linear changes in the resistance for various bending angles, from 0 to 80°. The corresponding change in resistance serves two purposes: (1) it improves the nonlinear range of the existing crack sensors, and (2) it quantitatively represents the amount of change in human motion so that the finger movement can be accurately monitored. Similarly, two V-groove cells arranged with a certain interval were demonstrated for monitoring finger joints at various axes (normal, 1 axis, and 2 axis), as shown in Figure 6b. The response of the crack sensor confirms that it is possible to effectively monitor the finger motion in multiple axes and easy to arrange according to the demands of various parts of the human body. Further, the crack sensor is suitable for monitoring movements with high degrees of freedom, such as multi-joint robotics, by arranging V-groove patterns of two or three cells.

To assess the sensor’s response performance, a test was conducted using a Morse signal. The Morse signal is a data transmission technique that sends and receives text messages based on the signal duration. In addition, the on-off indicators provided in the Morse code must be distinguished. As shown in Figure 7a, we used the proposed V-groove-induced crack-based strain sensor as a Morse code generator. The signal produced by the proposed V-groove-induced crack sensor was analyzed using the CNU MNTL text. The generation of Morse code determined that the reaction time between the signal and the next signal was around 99 ms, indicating a rapid response time, as shown in Figure 7b.

## 4. Conclusions

In this study, we came up with the V-groove-induced crack-based strain sensor to track human movement and make Morse code. The V-grooves were used to induce the direction and density of cracks in order to create a bio-inspired crack-based sensor. The crack-based sensor has precise control over the depth and shape of the crack, and it can be made using a simple imprinting technique. The proposed V-groove-induced crack-based sensor exhibited excellent fidelity and high deformation. The preliminary sensing characteristics showed that the proposed V-groove-induced strain sensor was more sensitive than the other reported metal-based strain sensors. Parallelizing the V-groove shape allows for multi-axis sensing, and studies have demonstrated that motion detection with botics, rapid feedback on motion, and a quick response time of 99 ms. The outcomes of the streamlined manufacturing procedure and sensing direction are anticipated to be implemented in the proposed V-groove-induced crack sensor in a diverse range of applications, including as a wearable sensor for human motion monitoring and next-generation AI convergence robotics.

## Figures and Tables

**Figure 1 micromachines-14-00062-f001:**
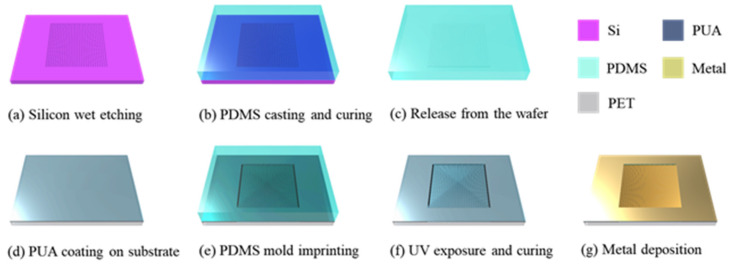
Schematic shows the fabrication process flow of the crack-based strain sensor realized through an UV curable polyurethane-acrylate coated film with V-groove arrays. (**a**) Silicon wafer wet etching (Etching depth 13 μm Negative V-groove shape). (**b**) PDMS casting and curing on the hotplate. (**c**) PDMS mold release from the silicon mold. (**d**) Photo sensitive material coating in the flexible polymer-based substrate. (**e**) V-groove pattern transfer onto a film substrate coated with a PUA material. (**f**) After imprinting contact, exposure UV light and V-groove mold remove from the substrate. (**g**) Metal thin layer was deposited on the V-groove based crack sensor whole area by using sputter system.

**Figure 2 micromachines-14-00062-f002:**
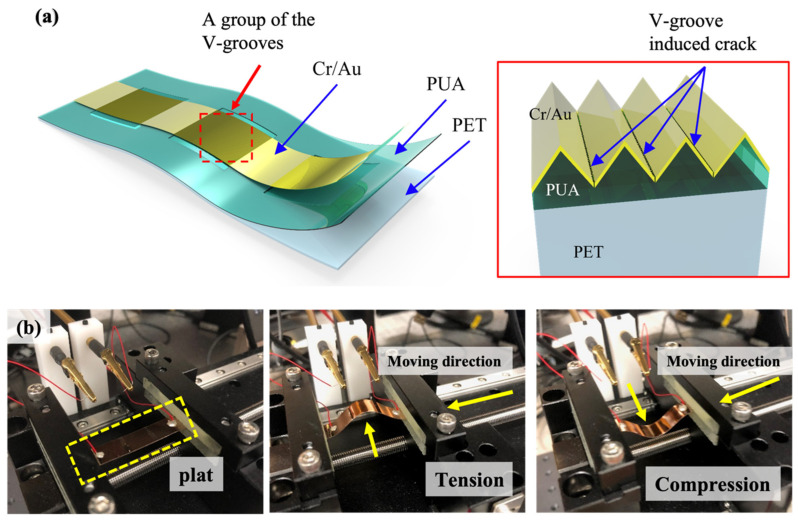
(**a**) Schematic illustration of the V-groove-induced crack-based strain sensor. (**b**) Photograph of the tensile testing machine used for the formation of regular cracks on the V-groove patterned PUA attached to the PET substrate.

**Figure 3 micromachines-14-00062-f003:**
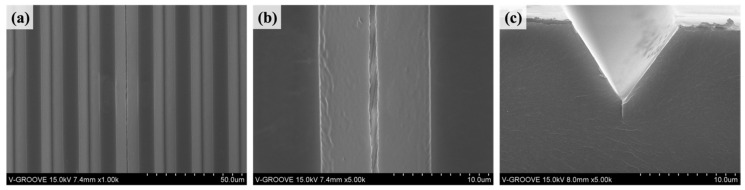
Field emission scanning electron microscopy images of the fabricated V-groove-induced crack-based strain sensor. (**a**,**b**) The panoramic view of the V-groove-induced crack-based strain sensor. (**b**,**c**) A highly magnified image shows the top view and cross-sectional image of the cracked V-groove-induced crack-based strain sensor under externally applied pressure.

**Figure 4 micromachines-14-00062-f004:**
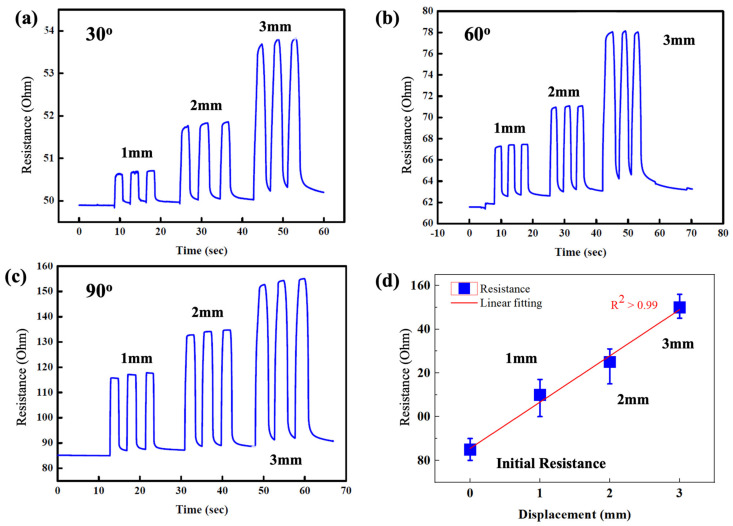
Preliminary sensing characteristics of the V-groove-induced crack-based strain sensor. (**a**–**c**) Effect of bending angles on the change in resistance of the V-groove-induced crack-based strain sensor at different applied displacements. (**d**) Average change in resistance of the V-groove-induced crack-based strain sensor at different applied displacements ranging from 1 to 3 mm at a 90° bending angle.

**Figure 5 micromachines-14-00062-f005:**
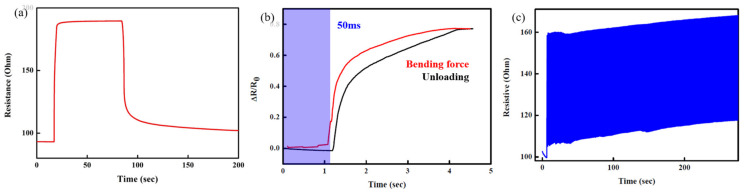
(**a**) Response and recovery time of the fabricated V-groove-induced crack sensor. (**b**) Hysteresis evaluation of the V-groove-induced crack sensor during loading and unloading of the external force. (**c**) Repeatability and long-term durability of the fabricated V-groove-induced crack sensor under the applied frequency of 3 Hz.

**Figure 6 micromachines-14-00062-f006:**
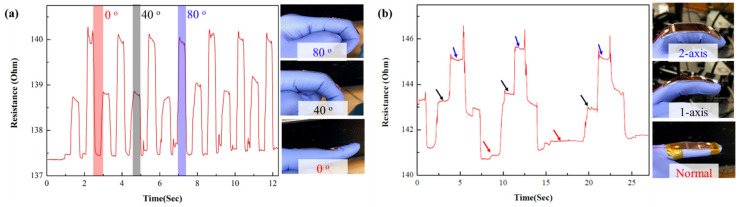
(**a**) Response evaluation of various finger bending motions at 0, 40, and 80°, respectively. (**b**) Resistance changes analysis of a 2 axis V-groove pattern array. The photograph shows the corresponding finger motions such as blue arrow: two axis, black arrow: single axis, and red arrow: normal state, respectively.

**Figure 7 micromachines-14-00062-f007:**
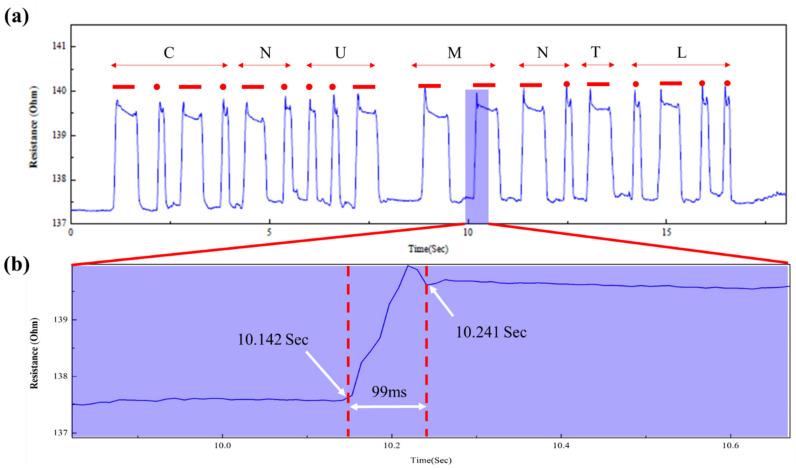
(**a**) Generation of Morse code using the proposed V-groove-induced crack-based strain sensor. (**b**) Response time of the V-groove-induced crack-based strain sensor during the generation of Morse code.

## Data Availability

Data will be available from the corresponding author upon reasonable request.

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
