# Peer review of "Crack-Based Sensor by Using the UV Curable Polyurethane-Acrylate Coated Film with V-Groove Arrays"

_micromachines, 2022, doi:10.3390/mi14010062_

Round 1

Reviewer 1 Report

This paper reports a detailed research concerning the fabrication and assessment of high durability and reproducibility strain sensors in detection of human motion and generation of Morse code using V-groove metal cracks array. The crack induced effect is clearly described and the results are discussed with competence. The obtained device could be really useful. The novelty of this work is the device, not the mechanism. However I believe and recommend this paper worthy for publication in the Micromachines.

Author Response

Response: We thank the Reviewer for their valuable time, appreciation, and very positive feedback on our work. The Reviewer’s words of encouragement surely boost us to continue our research efforts with a lot of confidence. We acknowledge the Reviewer’s valuable suggestions and recommending our manuscript for publication in this esteemed journal, “Micromachines”.

Reviewer 2 Report

In this manuscript, the authors reported a V-groove crack-based strain sensor with uniform cracks formed by concentrating stress along the groove. The concept of strain sensor is interesting as the customized cracks can improve the reproducibility of crack-based strain sensors, but some issues should be clarified before publication.

1. In page 5 (line 200 and 201), “the proposed V-groove induced crack sensor exhibits a wide operating range (0–10%)”. In this paper, the strain was applied by bending the sensor instead of traditional stretching, but the calculation method of the applied strain is unclear. I suggest the author add more details, such as the calculation method of the bending strain and the length of the sensor.

2. In page 5 (line 197 and 198), “The gauge factor of the fabricated crack sensor was calculated from the slope of the linear fit, and it was found to be 131.15”. Here, the graph of the linear fitting result should be given.

3. In page 5 (line 200), “but 10-fold lower than the reported crack-based sensor”. The “reported crack-based sensor” is unclear here. I suggest the author add the corresponding reference.

4. In page 4 (line 141), the unit of deposition rate (1 /s) may be miswritten. Please check and correct.

5. The method of inducing cracks by structure has been widely used in recent years. I suggest the authors to give a more comprehensive reference review in recent significant improvement in the field of strain sensors with structure-induced cracks (such as literatures in Adv. Mater. 2018, 30, 1704229, ACS Appl. Mater. Interfaces 2021, 13, 34637−34647, Adv. Mater. 2018, 30, 1706589, Nanoscale, 2018, 10, 15178–15186,4262, Sci. Adv. 8, eade0720 (2022)

Author Response

Response: We thank the Reviewer for their valuable time, appreciation, and very positive feedback on our work. The Reviewer’s words of encouragement surely boost us to continue our research efforts with a lot of confidence. We acknowledge the Reviewer’s valuable suggestions towards improving the quality of our manuscript. We modified the manuscript based on the Reviewer’s comments. The details of the changes made in the revised version of the manuscript are provided with this response letter for the Reviewer’s kind consideration. We sincerely hope the details provided along with the suggested modifications justify our conclusions and would help us to get an opportunity to publish in this esteemed journal, “Micromachines”.

Comment-1: In page 5 (line 200 and 201), “the proposed V-groove induced crack sensor exhibits a wide operating range (0–10%)”. In this paper, the strain was applied by bending the sensor instead of traditional stretching, but the calculation method of the applied strain is unclear. I suggest the author add more details, such as the calculation method of the bending strain and the length of the sensor.

Response: We sincerely thank the Reviewer for their valuable comment. As recommended by the Reviewer, we briefly described in detail the method that has been used for the calculation of strain values applied to the strain sensor. The strain applied to the sensor was calculated using equation reported in the previous manuscript [27]. We employed the bent radius and the central Θ factor when the tension strain applied on the sensor. The calculation method has been provided as supplementary information in the revised version of the manuscript.

Page no 5. Revised version of the manuscript.

The gauge factor of the fabricated crack sensor was calculated from the slope of the linear fit, and it was found to be 131.15 (Fig. 4d). The gauge factor of the proposed V-groove induced crack sensor was 10-fold higher than the gauge factor of the metal strain sensor, but 10-fold lower than the reported crack-based sensor [25]. However, the proposed V-groove induced crack sensor exhibits a wide operating range (0–10%) and excellent reliability at the base resistance. The strain applied to the sensor was calculated using equation reported in the previous manuscript [27] and described in the supplementary information.

Comment-2: In page 5 (line 197 and 198), “The gauge factor of the fabricated crack sensor was calculated from the slope of the linear fit, and it was found to be 131.15”. Here, the graph of the linear fitting result should be given.

Response: We sincerely regret our oversight, and it has been rectified in the revised version of the manuscript. We modified the manuscript based on the Reviewer’s comment and the suggested plot has been provided as the supplementary information in the revised version of the manuscript.

The applied stress significantly influences the depth and width of the cracks in the V-groove based strain sensor. As a result, we investigated the effect of bending stress on the resistance change of the V-groove induced crack-based strain sensor at various applied displacements of 1, 2, and 3 mm, respectively. The change resistance of the sensor was measured at different bending angles such as 30º, 60º, and 90º as shown in Fig. 4a-c. The initial resistance of the crack sensor increases with increasing bending angles. The base resistance of the crack sensor was found to be 50. 4 ± 0.3, 65. 3 ± 2.4, and 70. 3 ± 7.2 Ω toward 30º, 60º, and 90º, respectively. The increase in base resistance of the crack sensor with increasing the bending angle could be attributed to the formation of more depth and width of the cracks while applying the external bending stress. The formation of more cracks on the sensor decreased the contact point of the crack in the top metal layer, thus resulting in an increase in the initial resistance value. Although the base resistance of the crack sensor changed with respect to bending angles, the sensor greatly responded to the applied displacement as shown in Fig. 4a-c. The change in resistance of the crack sensor at 30º, 60º, and 90º to 3 mm of applied displacement was ~ 53.9 ± 1.1, 77.4 ± 2. 2 and 98.8 ± 2.4, respectively. The maximum change in resistance was observed at 90º due to the formation of more cracks in the V-groove induced crack sensor. The repeatability of the change in resistance of the crack sensor was measured at 90º bending angle toward 1, 2, and 3 mm, as shown in Fig. 4d. The crack sensor showed an excellent linear relationship to the applied displacement, demonstrating the excellent reliability of the proposed V-groove induced crack sensor. The gauge factor of the fabricated crack sensor was calculated from the slope of the linear fit, and it was found to be 131.15 (Fig. 4d). The gauge factor of the proposed V-groove induced crack sensor was 10-fold higher than the gauge factor of the metal strain sensor, but 10-fold lower than the reported crack-based sensor [25]. However, the proposed V-groove induced crack sensor exhibits a wide operating range (0–10%) and excellent reliability at the base resistance.

Comment-3:  In page 5 (line 200), “but 10-fold lower than the reported crack-based sensor”. The “reported crack-based sensor” is unclear here. I suggest the author add the corresponding reference.

Response: We sincerely regret our oversight, and it has been rectified in the revised version of the manuscript. As recommended by the Reviewer, we have provided the suitable reference in the suggested place.

Page no 5: Revised version of the manuscript.

The gauge factor of the proposed V-groove induced crack sensor was 10-fold higher than the gauge factor of the metal strain sensor, but 10-fold lower than the reported crack-based sensor [25]. However, the proposed V-groove induced crack sensor exhibits a wide operating range (0–10%) and excellent reliability at the base resistance.

  1. Kim, J.; Lee, M.; Shim, H. J.; Ghaffari, R.; Cho, H. R.; Son, D.; Jung, Y. H.; Soh, M.; Choi, C.; Jung, S.; Chu, K.; Jeon, D.; Lee, S.-J.; Kim, J. H.; Choi, S. H.; Hyun, T.; Kim, D.-H. Stretchable silicon nanoribbon electronics for skin prosthesis. Commun. 2014, 5, 5747.

Comment-4:  In page 4 (line 141), the unit of deposition rate (1 /s) may be miswritten. Please check and correct.

Response: We sincerely regret our oversight, and it has been rectified in the revised version of the manuscript.

Page no 4: Revised version of the manuscript.

Finally, using an electron beam evaporator at a deposition rate of 1 nm s-1 and a chamber pressure of less than 5 10-6 torr, 30 nm chromium (Cr) and 30 nm gold (Au) layers were deposited on the V-groove pattern PUA substrate.

Comment-5:  The method of inducing cracks by structure has been widely used in recent years. I suggest the authors to give a more comprehensive reference review in recent significant improvement in the field of strain sensors with structure-induced cracks (such as literatures in Adv. Mater. 2018, 30, 1704229, ACS Appl. Mater. Interfaces 2021, 13, 34637−34647, Adv. Mater. 2018, 30, 1706589, Nanoscale, 2018, 10, 15178–15186,4262, Sci. Adv. 8, eade0720 (2022).

Response: We sincerely thank the reviewer for providing us the useful references. Based on the Reviewer’s recommendation, we modified the part of the introduction in the revised version of the manuscript.

Page no 2: Revised version of the manuscript.

Recently, a new biomimetics sensing mechanism with an effective design and high sensitivity for providing high sensing performances was developed. The crack-based sensor possesses high sensitivity even for small displacement or voice vibration [25]. The bioinspired crack-based sensors apply a strain of about 2% to create micro- or nanoscale physical cracks in both electrical metals and flexible substrates [26, 27]. For instance, Liu et al. proposed new strategy to develop fiber shaped stretchable strain sensors with significantly improved sensitivity [28]. Chen et al reported highly sensitive and fully flexible vibration sensor with a channel-crack-designed suspended sensing membrane for high dynamic vibration and acceleration monitoring [29]. Jiang et al. proposed the stretchable strain sensors based on auxetic mechanical metamaterials. The proposed crack sensor showed 24-fold sensitivity improvement compared to the conventional sensors [30]. The high-performance flexible strain sensor with bio-inspired crack arrays has been proposed for detection of human motions and surface folding [31]. Li et al. proposed bioinspired robot skin with mechanically gated electron channels for sliding tactile perception [32]. The crack sensor has a high gauge factor of between 11 and 2000 and can monitor even small vibrations or sound waves. Though the crack-based sensor has higher sensitivity compared to that of a piezoresistive sensor, it does not have a uniform primary resistance for each device as the number of cracks cannot be controlled. Furthermore, the majority of crack-based sensors operate within a 2% tension range, with the crack creating extending ranges on the sensor surface [33]. To overcome these drawbacks, the researchers developed advanced fabrication methods for the crack-based sensor and extended the application fields. Furthermore, special equipment was required to make the crack, and still, it did not have uniform initial resistance and sensitivity.

  1. Liu, Z.; Qi, D.; Hu, G.; Wang, H.; Jiang, Y.; Chen, G.; Luo, Y.; Loh, X. J.; Liedberg, B.; Chen, X. Surface strain redistribution on structured microfibers to enhance sensitivity of fiber-shaped stretchable strain sensors, Mater. 2018, 30, 1704229.
  2. Chen,; Zeng, Q.; Shao, J.; Li, S.; Li, X.; Tian, H.; Liu, G.; Nie, B.; Luo, Y. Channel-crack-designed suspended sensing membrane as a fully flexible vibration sensor with high sensitivity and dynamic range, ACS Appl. Mater. Interfaces. 2021, 13, 34637−34647.
  3. Jiang,; Liu, Z.; Matsuhisa, N.; Qi, D.; Leow, W. R.; Yang, H.; Yu, J.; Chen, G.; Liu, Y.; Wan, C.; Liu, Z.; Chen, X. Auxetic mechanical metamaterials to enhance sensitivity of stretchable strain sensors, Adv. Mater.2018, 30, 1706589.
  4. Han, ; Liu, L.; Zhang, J.; Han, Q.; Wang, K.; Song, H.; Wang, Z.; Jiao, Z.; Niu, S.; Ren, L. High-performance flexible strain sensor with bio-inspired crack arrays, Nanoscale, 2018, 10, 15178–15186.
  5. Li,; Chen, X.; Li, X.; Tian, H.; Wang, C.; Nie, B.; He, J.; Shao, J. Bioinspired robot skin with mechanically gated electron channels for sliding tactile perception, Sci. Adv. 2022, 8, eade0720.
